# Understanding Hand Hygiene Behavior in a Public Hospital in Benin Using the Theoretical Domain Frameworks: The First Step for Designing Appropriate Interventions

**DOI:** 10.3390/healthcare10101924

**Published:** 2022-09-30

**Authors:** Carine Laurence Yehouenou, Aynaz Abedinzadeh, Roch Houngnihin, Carine Baxerres, Francis M. Dossou, Anne Simon, Olivia Dalleur

**Affiliations:** 1Clinical Pharmacy Research Group (CLIP), Louvain Drug Research Institute (LDRI), Université Catholique de Louvain, 1200 Brussels, Belgium; 2Laboratoire de Référence des Mycobactéries (LRM), Cotonou BP 817, Benin; 3Faculté des Sciences de la Santé (FSS), Université d’Abomey Calavi (UAC), Cotonou 01 BP 526, Benin; 4Service de Pharmacie Clinique, Clinique Universitaires Saint-Luc, UCLouvain, 1200 Brussels, Belgium; 5Laboratoire d’Anthropologie Médicale Appliquée (LAMA), Université d’Abomey-Calavi, Cotonou 01 BP 188, Benin; 6UMR261-MERIT, French National Research Institute for Sustainable Development (IRD), University of Paris, 75006 Paris, France; 7UMR151-LPED, IRD, Aix Marseille Université, 13005 Marseille, France; 8Department of Surgery and Surgical Specialties, Faculty of Health Sciences, Campus Universitaire, Champs de Foire, Cotonou 01 BP 118, Benin; 9Centres Hospitaliers Jolimont, Prévention et Contrôle des Infections, Groupe Jolimont Asbl, Rue Ferrer, 7100 Haine-Saint-Paul, Belgium

**Keywords:** hand hygiene compliance, healthcare workers, implementation science, theoretical domains framework, qualitative interviews, Benin

## Abstract

Background: Hand Hygiene (HH) is widely recognized to be one of the most successful and cost-effective measures for reducing the incidence of healthcare-associated infections (HAIs). The hand hygiene behavior of hospital healthcare workers (HCWs) is not well-documented in Benin. Therefore, Theoretical Domains Framework (TDF) was used to identify the behavioral determinants that may impact HCWs’ hand-hygiene compliance in a public hospital. Methods: A qualitative design comprising face-to-face semi-structured interviews with nine HCWs. The interviews included questions on transmission of infections, hand-hygiene practices, problems with their implementation; and ways to improve hand hygiene compliance. Two pharmacists independently coded interviews into behavioral domains using the TDF and then subdivided them into several themes. Interview transcripts were analyzed following 3-steps approach: coding, generation of specific beliefs, and identification of relevant domains. Results: Almost all interviewees have cited the environmental context and resources (such as lack of water) as a barrier to HH practice. They also believed that role models had a significant impact on the good practices of others HCWs. Fortunately, they were confident of their capabilities to perform appropriate HH behaviors. The majority (7/9) reported having the necessary knowledge and skills and believed they could carry out appropriate HH behavior. In all cases, the participants were motivated to carry out HH behavior, and it was recognized that HH remains the cornerstone to reduce health care associated infections. Conclusion: This study identified several behavioral constructs aligned with the TDF that can be targeted and help for the development of new hand-hygiene interventions. These may increase the likelihood of a successful intervention, thereby improving HH compliance and patient safety, especially in hospitals.

## 1. Background

Evidence shows that effective and consistent hand-hygiene practices have proven their effectiveness as simple but important measures for prevention of the healthcare-associated infections (HAIs) [1]. Sustained improvements in hand-hygiene practices were therefore a challenge for the World Health Organization’s (WHO), whose intention was to help in decreasing of the international burden of HAIs [2]. However, the compliance of healthcare workers (HCWs) with these practices is poor [3], as recently shown in Benin [4]. 

HCWs’ compliance with hand hygiene guidelines has generally been determined using self-reporting, observational, or interventional approaches. The common barriers described with these methods were: the sex (being female), being a physician rather than a nurse, working in an intensive care unit, working during weekdays rather than the weekend, lack of time and organizational support [5]. Nevertheless, direct observation studies may be biased by observer effects (also known as Hawthorne effects) [6]. Similarly, data collection using self-recall method may suffer from memory-recall bias [7]. For these reasons, data collected using qualitative methods may be valuable interviews can provide rich data that better explore the issues under examination, such as HCWs’ compliance with HH guidelines [6]. Qualitative studies are recognized as making deeper contributions to evidence-based practice and health services-research [8]. Moreover, interviews can help researchers to access the thoughts and feelings of research participants. 

Various factors have been associated with poor HH practices in healthcare settings, including a lack of infrastructures (number and location) and limited behavioral change interventions [9]. Indeed, to successfully design a hand-hygiene promotion intervention, we first need to understand the determinants of current behaviors informed by a theory of behavioral change. The reasons underlying HCWs’ low levels of hand-hygiene compliance in resource-limited settings are rarely investigated, and little is known about current beliefs either on the advantages of infection control strategies, or on behavioral enablers for infection control improvement. 

Different theories of behavior change exist, but with similar and overlapping constructs [10]. The diversity and number of these theories have been identified as possible reasons why theoretical approaches are not used in the design of interventions. The (TDF) was developed to provide a structure to support the application of theoretical approaches to interventions with the goal of contributing to the change of behavior. The TDF is a well-validated, predefined, consensus-based, theoretical framework for supporting the implementation of healthcare guidelines and consist of 14 domains as follows: knowledge, skills, social professional role and identity, beliefs about capabilities, optimism, beliefs about consequences, reinforcement, intention, goals, memory attention and decision processes, environmental context and resources, social influences, emotion, and behavioral regulation. [11,12]. By providing a common language for characterizing contexts [13], targeted problems and interventions in generalizable terms, it also guides the adaptation of implementation strategies [14]. 

Therefore, the purpose of the study was firstly to use the TDF as an analytical framework to gain a global understanding of HH behavior in public hospital in Benin through semi-structured interviews with healthcare workers. Secondly, according to the analysis of these interviews, suggestions for the types of interventions that may be effective in improving the HH compliance of HCWs will be briefly described. In this way, we propose to diagnose the barriers and enablers to HH compliance in hospital, and thereby prescribe appropriate interventions to address their local challenges in a targeted way.

## 2. Methods

### 2.1. Study Design

This descriptive qualitative study used semi-structured interviews to explore HH compliance based on participants’ first-hand accounts of their experiences. This study was part of the MUSTPIC (Multidisciplinary Strategy for Prevention and Infection Control) study that assessed hand hygiene practices and promote rational use of antibiotics in surgery services in Benin. They are different activities among which we observed hand hygiene practices among healthcare workers, analyze the quality of antibiotics used for prophylaxis and described the epidemiology of multidrug resistant bacteria involved in surgical site infections [15]. We report here qualitative results based semi-structured interviews. 

### 2.2. Setting 

The study was conducted between February and March 2019. Participants were HCWs at a 36-bed confessional hospital directed by clergymen in Cotonou. This hospital has the cheapest consultation fees of public and private hospitals in Benin. Based on our purposive sampling method, we sought to include caregivers working at critical point of care, such as surgery. To ensure a maximum level of participation, the interviews took place on the surgical ward at places and times convenient to the interviewees.

### 2.3. Participants

Eligible participants were physicians, nurses, surgeons, and cleaning staff in the surgery ward. In surgery department we have count 23 healthcare workers for all specialties as follows: 5 cleaning staff members, 10 nurses, 4 surgeons and 4 physicians. All were aged 18 or older, were in regular physical contact with patients, and would not be on vacation or an extended period of leave when the interviews were conducted. This study is integrated on MUSTPIC project which aimed to describe hand hygiene compliance and understand the perceptions of HCWs about hand hygiene as well as the perspectives for improvement. 

### 2.4. Data Collection

The interviews were carried out by three facilitators (2 medical psychologists) who did not know the participants they interviewed, while a second one took notes and made audio recordings. Interviews were conducted at a place and time chosen by the participants. These interviews were download by Olympus DSS player software to laptop to be analyzed. Then, interviews were transcribed verbatim. The question grid included open-ended questions, which allowed new ideas and questions to arise out of the participants’ responses. Each session began with a small introduction in which a participant met the researchers, was given a brief description of the study’s goals, and an assurance of confidentiality. Before beginning each session, each participant provided oral informed consent. No monetary incentive was offered. 

### 2.5. Data Analysis

To ensure rigor and trustworthiness of the data, content analysis of the interview transcript was performed by 2 researchers (CY, and AA) who did not take part in recruitment or not conducting the interviews. No software was used to support analysis. Analysis was performed through annotated copies of the interviews. After coding the interviews independently, the two coders gathered, compared their codes, and resolves disagreements through discussion to achieve consensus. Following completion of analysis, exemplar quotes for each domains were chosen by consensus between the two researchers. 

The data was analyzed on the basis of the TDF, which we had chosen because it integrates behavior-change theories and uses tools that support the implementation of behavior-change interventions [16]. The TDF framework was initially developed for implementation research to identify influences on health professional behavior related to implementation of evidence-based recommendations. A synthesis of 33 theories of behavior and behavior changes clustered into 14 (originally 12) domains [17], the TDF is a theoretical framework rather than a theory and it provides a theoretical lens through which to view the cognitive, affective, social, and environmental influences on behavior [18]. The 14 domains provide a complete coverage of the potential multi-level determinants of health-related behavior and guide the use of broad prompts that enable interviewees to consider a wide range of possibilities without asking leading questions [19]. As a reminder: domains are simplified groupings of constructs from multiple theories, and constructs are the components of a theory used to explain behavior [16]. 

The recordings were verified by the interviewer prior to analysis, which was facilitated using Microsoft Word and consisted of the two following steps: 

Coding. To facilitate analysis and ensure consistency in coding, the transcripts were coded into TDF domains by two team members trained in qualitative analysis using thematic content analysis. Beforehand, however, a coding scheme was determined by consensus by two team members who had read the first 2 transcripts. This scheme comprised codes provided definitions and examples of the codes and provided examples of quotes that were covered by the codes. The code resulting from this process was then used to analyze remaining transcripts. 

Generation of specific beliefs. A specific belief is a collection of participant responses regarding theme that suggests a problem and/or influence on the target behavior. For each utterance (i.e., for each coded interview quote), specific beliefs were generated in TDF domains by one team member and double-checked for accuracy by a second team member. Beliefs statements were initially written to be specific to each code; later, similar belief statements were merged to form the themes.

## 3. Results

### 3.1. Characteristics of Interviewees

A total of nine participants (five male and four female) were recruited for the interviews; their characteristics are summarized in Table 1. 

The time they had worked on the surgical ward ranged from 4 to 11 years. The interviews duration ranged from 45 to 60 min. The analysis identified 11 theoretical domains. (Table 2). 

We identified eight behavioral themes that influenced participants’ HH compliance. These themes [20] were mapped against the theoretical domains (Table 2). Further analysis allowed them to be classified either as enablers of correct HH or as barriers against it. Four enablers and four barriers were identified: As enablers the concerned domains were (a) their beliefs about their socio-professional role as HCWs, (b) their knowledge of hand hygiene, (c) their social influences, and (d) beliefs about the negative consequences of poor HH. 

The identified barriers were: (a) Memory attention and decision process (i.e., participants’ understanding of the importance of being a role model to their colleagues), (b) behavioral regulation, (c) environmental resources (such as a shortage of supplies and the unavailability of clean water) and (d) participants’ beliefs about negative consequences [16] to themselves (such as skin irritation). Almost half of the responses on the environmental resources theme were related to the unavailability of clean water. The remaining half were attributed to the bad quality of the only water that was available. As a barrier to HH, damage to skin [16] was the predominant concern of half of the participants; it had been coded to the theme of beliefs about negative consequences to self. 

### 3.2. Enablers Factors 

#### Social Professional Role and Identity

An important enabler of optimal HH compliance was the professional responsibility [20] some physicians felt towards protecting they needed to treat patients from infections which would be the exact opposite of the initial intention. For example, one physician said, “HCWs care about HH in hospital, healthcare is a primordial gesture” (interview 2).

Other’s participants said that protecting themselves and their family had a greater priority than protecting their patients. Protection for themselves and their family was reported to be strong motivators of HH compliance: “*The aim of hand-washing is to protect the patients, but mostly to protect myself and my family as well*” (interview 4, surgeon). 

Some participants believed that, as much as needing to refresh HH practice and usefulness and educational strategies, the role of a model was important to reminding colleagues about the WHO’s five moments as followed: (a) before touching a patient, (b) before clean/aseptic procedure, (c) after body fluid exposure, (d) after touching a patient and (e) after touching patient surroundings [1]. Some interviewees appear to copy the HH behavior of the physicians they see at work, often resulting in poor HH habits that will, in turn, be copied by future healthcare workers (HCWs). Interviewees insisted on patients’ sensibilization as they can also prevent the transmission of HAIs *“It was necessary to raise awareness and put posters at the water points. At the very least, these attract visitors’ attention, and make them look”* (interview 3, nurse). Posters are one of the solutions our participants proposed. 

### 3.3. Knowledge 

Knowledge of hand hygiene procedures was an important enabler of optimal HH compliance among interviewees [16]. However, despite the knowledge of procedures and the evidence linking HH to healthcare-associated infections, HH technics were not always performed correctly. They were aware of the advantages of good HH and recognized that they were sometimes superficial during their daily tasks. As a surgeon noted, *“when I wash my hands, I create a kind of wall between myself and the patient. So bacteria are on one side of the wall, and the patient is on the other side”* (Interview 4, surgeon). 

*“You cannot touch a suppurated wound and only rub after your hands afterwards. Normally you must wash your hands beforehand”* (interview 4, surgeon).

Although most participants have sufficient knowledge about the practice of HH, some professional groups needed to improve their practices of the proper technique, and to use it at the right time. As one participant noted, *“I wash my hands every time, but I’m not sure I always do it correctly?”* (Interview 3, nurse). 

Some participants highlighted the important problem of patient empowerment in the hospital’s HH practices, and that patients and visitors had also contributed to improvements in HH: *“While staff working in the hospital are aware of HH guidelines, those who come occasionally have no idea about them, which is ignorance”* (interview 2, physician).

One important point is knowledge of the benefits of handwashing. Medical personnel are aware of its advantages, and often say that it is this knowledge that causes them to be compliant. *“It’s because I’m a health worker that I know all the benefits of handwashing”* (interview 1, nurse). 

### 3.4. Social Influences 

Participants indicated that their colleagues were supportive about HH practices, and that they could count on each other to improve HH, although one nurse confided to us that when she respected properly hand hygiene guidelines all the time, she was treated like a “figure of fun” by other nurses. On the other hand, HH practices sometimes became an obsession for HCWs. Since—as stated above—they were conscious of the consequences of not respecting it, they could not prevent themselves from practicing it. *“I already know that germs are always present in the hospital, and that’s how it is in my head”* (interview 1, nurse). 

### 3.5. Beliefs about the Negative Consequences of Poor Hand Hygiene

All the participants expressed a belief that lack of HH contributes to increasing of infection and poor patient outcomes: *“The lack of HH could harm somebody and provoke supplementary costs for patients […] when infections appear, we use higher doses of antibiotics”*. (Interview 3, nurse). 

HCWs also recognized the impact of their own forgetfulness and negligence during their daily tasks. *“When it is respected correctly, hand hygiene reduces most germs, and avoids healthcare-associated infections”* (interview 3, nurse). 

They were also aware that forgetfulness and negligence could be a source of illnesses they themselves could contract—another reason to pay attention to it: *“[HH] is a good thing: it prevents disease”* (interview 5, cleaning staff).

### 3.6. Barriers to Proper Hand Hygiene

#### 3.6.1. Memory Attention and Decision Process

Most participants (6/9) expressed a strong belief that the basic approach of education, reminders and training had had a considerable effect. But the content, focus, the frequency, and modes of delivery needed to be modulated and refreshed. One participant talked about the important role played by the model of prevention and control of infection in the hospital. He had to remind other colleagues of the WHO five moments and the importance of HH: *“Sometimes, I would like to talk about ignorance, it is ignorance for the visitors, we should put on all waters source reminders and HH posters”* (interview 3, nurse).

Although, the other members of team could be counted on give reminders and improve compliance, they were sometimes simply inattentive and forget to wash their hands. *“To be honest, if I want to go to the patients, I may sometimes forget to wash my hands, but when I come back, I remember to wash them ”* (interview 3, nurse).

In hospitals, time is an important consideration. Staff are sometimes overwhelmed and do not take the time to sanitize their hands properly when they should. *“If I didn’t have time to go out to sanitize them during the operation, I will do it before I finish with the patient at the end of the operation”* (interview 1, nurse).

#### 3.6.2. Behavioral Regulation 

All interviewees believed that periodic or continuous training by hospital authorities should be promoted at specific times and should be contain posters and specific hand hygiene techniques. A member of the cleaning staff believed that *“one of the most important things we’ve done was the training and awareness-raising”* (interview 5, cleaner). Nonetheless, some of (3/9) participants stated that they had never been trained in HH. 

According to our participants, hand hygiene was above all a question of nail hygiene, and that, to meet hospital standards, cutting and/or properly cleaning your nails was therefore an essential part of complete hand hygiene. Usually, most of them agreed with the following statement: *“the nails hide germs inside for those that let the nails grow and do not maintain them” (interview 1, nurse).* Nurses, particularly the nurse in interview 1, were more likely to describe HH as habitual behavior and recognized that they had received at least one training program on it. This point is illustrated by the following quote: *“Although we receive hand-hygiene training all of the time, we also need to know the practical aspects, not just the theoretical ones.”* (Interviews 2).

#### 3.6.3. Environmental Context and Resources 

When participants were asked about what most influenced their HH practice, they referred to two things: organizational culture and the availability of resources. System constraints were consistently identified as important barriers to HH compliance. If HH resources were easily available within the clinical environment, they triggered hand hygiene. *“If the sinks were better located, I would perform HH better”* (interview 8, surgeon).

The lack of alcohol-based hand rub (ABHR) and sinks in consultation rooms, the chaotic context was all reported to be barriers to good hand-hygiene compliance. The availability of soap and ABHR only one time out of two was a barrier to the convenient use of soap or alcohol for hand hygiene. *“The ABHR is not compulsory—I mean, it’s not always available”* (interview 4, surgeon).

Some interviewees described more undesirable quality of water on the hospital and said they were afraid to use it to wash their hands. *“We’re in Benin here, in Cotonou. Haven’t you ever seen dirty tap water?”*(interview 3, nurse). Due to that fear, they brought their own water: “ *I don’t entrust my health to the hospital; if I’m not well, I can’t..., I can buy water outside, and keep that to wash my hands with”* (interview 1, nurse). 

The hospital also lacked taps whose locations were inconvenient: “I think there are taps, but there aren’t many of them, and they’re not located properly (interview 1, nurse). “Sometimes the water or the tap doesn’t work in surgical area, and surgeons are forced to wait, or find water in other services” (interview 5, cleaning staff). As well as the lack of taps or their poor localization, there was another obstacle to compliance: water cuts. “Sometimes the water is turned off. At your home, do you turn off the water?” (Interview 8, surgeon). 

Some participants—particularly cleaning staff—highlighted the fact that they were forced to make water provision in reservoir (see interviews 5 and 8). 

### 3.7. Beliefs about Negatives Consequences of Practicing HH

Some interviewees have challenged us about how dry they hands became because of HH practice in work. They believed that the consequences on the quality of their skins constituted a sort of barrier to their hand hygiene on a daily basis. As you can see. The “belief about negatives consequences” can be benefit on one hand and a barrier on the other hand. 

## 4. Discussion

This qualitative study explored determinants of successful HH practice in healthcare professionals in Benin. Our participants identified many elements that commonly affect the compliance of their colleagues around the world. Our study is the first to use the Theoretical Domain Framework to systematically understand enablers and barriers of HH in Benin. To identify the specific determinants of performance of HH, and improve it, behavioral research into HH is needed to design tailored interventions. Indeed, these factors—which range from personal beliefs to organizational and social contexts—are potential targets for behavior-change interventions that guide improvements of HH compliance. Using TDF, we identified 11 domains representing potential barriers to, or enablers of HCWs’ hand-hygiene practice. At the heart of our findings were six domains that were the most likely impact adherence to HH guidelines: (1) environmental context and resources, (2) social/professional role and identity, (3) knowledge, (4) beliefs about consequences, (5) behavioral regulation and (6) memory attention and decision process. 

We found that, as key enablers of optimal HH practice, participant’s knowledge and skills were interconnected. Some participants reporting having received some instructions on HH during their professional or induction training. As knowledge of HH is often reported to be one of the most important determinants of actual HH behavior [21,22], training is an important factor in behavioral improvement. All the interviewees in our study did indeed report that their beliefs in the “knowledge” domain had a big effect on their HH behavior. They were aware of the importance of HH in reducing healthcare-associated infections and recognized the benefits of good HH adherence. However, even though training and education have been claimed to play an important role in improving HH compliance and are supported elsewhere as a pivotal influencer [23,24], some interviewees noted the theoretical aspects of various HH courses. They suggested that, until high compliance was reached, hospital administrators should continue providing HH training based on observation and immediate feedback. There was also a need to change this training model based on a structured behavior modification program. 

While most of the HCWs defined hands as the major vehicle for the transmission of infections, the two cleaners we interviewed were still uncertain about when to perform HH. Their assessment of the need to perform HH was influenced by the concepts of “cleanliness” and “dirtiness,” which they perceived or sensed emotionally, feelings of “dirtiness” being evoked particularly by intimate contact with patients and bodily fluids [25]. In interviews, they also made a distinction between something that was clean and something that was sterile. Even if appearance is already a good indicator of cleanliness, it cannot always show whether something is dirty. The infrequent cleaning and poor ward hygiene was highlighted in many other studies in low middle-income countries where HCWs emphasized the influence of resource constraints and needed inputs [26,27]. They also pointed something they thought important about HH in the hand itself: the nails. For them, nails were a good vector of infections. For HH to be performed properly, they proposed that the nails should be cut or cleaned with special attention. 

Another most cited barriers to good HH adherence were environmental context and resources (mainly lack of time and accessibility of products). When the workload was high, HCWs’ compliance was affected by the ease or difficulty of access to HH products. In our institution, alcohol-based products are located one case out of three [4]. Some authors showed that the conviction that HH required relatively little effort was consistently associated with good adherence [28]. Access but also the proximity of resources was discussed, by sax [28]. Thus, for example, some interviewees (5/9) needed the waters points to be located on each floor of the hospital. 

Role models remains a significant key for good practice of the HCWs interviewed. The potential for senior doctors or infection prevention and control (IPC) members as role models has been frequently identified in the literature [29,30]. Not only do behavioral models consider role models to be a significant part of the decision-making process [31], but compliance has also been found to be greater in those who perceive themselves to be role models [32]. In contrast, Lankford et al. demonstrated that HCWs were negatively influenced about HH if they were in room with a peer or higher-ranking person who did not perform it [33]. Even if these findings suggest that HH behaviors can be affected by peer or role-model HH compliance, group compliance with HH procedures may be negatively influenced by learned behaviors or time constraints [33]. 

Participant’s understanding and abilities about improving HH compliance were consistent with their professional responsibilities. Another enabler of optimal HH compliance was the professional responsibility; some physicians were concerned about the protection of patients from infections. The influence of the professional group of a HCWs is also important. While all physicians in our study reported to correctly perform HH, recent studies indicates that physicians are often excluded from studies of HH compliance, and showed underperformance compared to their colleagues in nursing and allied health professions [29]. Moreover, in our setting, hand hygiene topic is not included in training curriculum. Protecting themselves and not taking infections home to their family were reported to be strong enabler to effective HH. As HCWs consider themselves as an important intermediary between patients and especially their illnesses, they see themselves as a good vector of transmission from patient A to patient B. To evaluate explanations for non-compliance to HH best practices in real-time, Fuller and colleagues used a codebook based on the TDF [34] and reported that 44% of explanations for non-compliance with HH could be mapped to attention memory, and decision-making. In our study, this domain represents the third most common barrier.

As part of an overall strategy for facilitating patients’ active patients ‘active involvement in their healthcare management, it has also been suggested that patients and visitors be empowered to remind staff to wash their hands [35,36], even though patient empowerment has more commonly been used in relation to chronic disease management than in acute care settings [37]. In Australia, authors have actively promoted empowerment patients’ and visitors through HH campaigns such as “it’s OK to ask,” or by reminding hospital staff to wash their hands [38]. However, while some clinical managers were supportive of trialing patient empowerment to remind staff to wash their hands, they also identified several potential barriers to implementing this strategy. 

All in all, HCWs’ compliance with HH continues to be a major challenge. Ongoing education and information are necessary for HCWs, patients and visitors alike. Our own HCWs’ key recommendations for supporting HH compliance focused on (a) the availability of environmental resources such as hand-care products and facilities for performing HH in convenient locations; (b) awareness-raising (patients and HCWs); and (c) training in the practical aspects of HH. 

The strengths of this study lie first in the use of the TDF—to help in the establishment of the process of implementation [39]. Secondly, by interviewing physicians and nurses, we gained insight into the perspectives of the key groups responsible for most inpatient medical services in the hospital. A potential weakness is the risk of response bias by participants who could have been influenced by a desire to provide the “correct response”. Another limitation is the relatively low number of interviewees. However, even though our interviewees differed about their roles and specialties, their responses to the question of HH tended to converge, indicating that the behavioral determinants were similar. We also had the opinion of other workers such as cleaners. Even though, at first glance, they did not have the same knowledge of the pathologies, or the risks related to non-compliance, their opinions were like those of the other interviewees. To avoid the biases related to objectivity that may be present in these types of studies, our study was also carried out independently by two investigators who then pooled their work. This qualitative study represents a backbone for further interventions in hospitals in Benin. 

## 5. Conclusions

Best practice for improving hand hygiene in hospitals has not yet been established, and compliance in developing countries remains non optimal. Although direct observation is considered as the “gold standard” method for the measurement of HH compliance, it provides little insight into why a particular behavior does or does not occur. Viewed from the perspective of HCWs, several suggestions can be made for strengthening hospital hand-hygiene strategies. The first involves organizational support and leadership, which should include role modelling. The second involves ongoing education of HCWs, visitor and patients on the importance of good HH compliance. This study is the first step that come before implementation of a solid system change in our healthcare setting. 

## Figures and Tables

**Table 1 healthcare-10-01924-t001:** Characteristics of interviewees.

Interviewees	Number (n)
Gender	
Male	05
Female	04
Staff profession	
Surgeons	02
Physicians	02
Nurses	03
Cleaning staff	02

**Table 2 healthcare-10-01924-t002:** Resume of TDF and illustrative excerpts.

Theoretical Domains Frameworks	Themes Aligned	Illustrative Excerpts
Knowledge	Awareness of HH guidelines and advantages Knowledge of HH policy and procedures Awareness of evidence linking HH to healthcare-associated infections Good understanding of advantages of HH	“*When I wash my hands, I create a kind of wall between 2 patients. So bacteria are on one side and patient on the other side of wall*” (interview 4, surgeon).
Skills	Completing HH training by technical aspect Frequent repetition of HH training by themselves Infection control education and communication Regarding HH as a skill	“*We need a training more and more, moreover and these trainings sessions need to contain more than theoretical information –the HH technic is also important*” (Interview 3, nurse).
Social/professional role and identity	HH should be completed correctly by all caregivers HH should be performed not only by caregivers but also patients and visitors Importance of patients’ role in avoiding the transmission of infection	“*Importance of hygiene of the nurses who are also in first contact with the patients, and the hygiene of the nursing staff, doctors and all those who pass by*” (Interview 2, physician)
Beliefs about capabilities	Confidence that HCWs will follow HH guidelines if HH infrastructures are improved Being confident and positive about improving HH compliance	“*This hospital can be better if they give us resources and infrastructures*” (Interview 2, physician). “*It would be a lie if I said that this hospital was totally clean. I can’t declare that the level of cleanness is high. I give it five out of ten—the hospital is not totally clean*” (Interview 7, cleaner)
Beliefs about consequences	Performing HH reduces the rates of healthcare associated infections Performing HH damages my hands, particularly ABHR, which makes them dry	“*Sometimes I wash my hands, but without efficacity because we have doubt about the quality of water. Efficacity of HH depends on technique too*” (Interview 1, nurse).
Motivations and goals	HH is always a necessity The importance of HH in self-protection HH knowledge and training are a necessity	“*Proper hand hygiene is effective at protecting my family, my colleagues and patient lives, so it firstly to protect my own self*” (Interview 9)
Memory attention and decision process	HH posters are useful for my HH daily practice Practicing HH is a habit	“*We would do better with a model, someone who reminds us of the HH practices and guidelines*” (Interview 8, surgeon).
Environmental context and resources	Easy access to hand-hygiene products makes it easier to practice HH The quality of water is very important for HH to be effective Sometimes HH is not effective and suffers from the lack of materials	“*The difficulty is the great lack of infrastructure in our hospital. And sometimes the quality of water is also doubtful*” (Interview 1, nurse) “*It’s difficult to measure the quality of hand washing… they just wash their hands, and think it’s clean enough*” (Interview 3, nurse) “*if the sink would be better located, I will perform better hand-hygiene action*” (Interview 2, physician)
Social influence	The model can influence HH practice Leadership and role models on the ward can help us to improve HH compliance.	“*We can count on the team to improve HH compliance—they will remind their colleagues to wash their hands*” (Interview 4, surgeon)
Emotion	The feeling of dryness and irritability of skin caused by recurrent washing A kind of security procured by hand washing	“*I prefer washing with soap to the alcohol use because of dryness that it occurs*” (Interview 4, surgeon)
Behavioral regulation	Education of patients and continue frequently the training of HCWs Information and sensibilization Ability to prioritize and organize goals Hand-hygiene audits	“*Education, information and consciousness-raising concerns to all of us: patients, HCWs and visitors*” (Interview 8, surgeon).

## Data Availability

The data supporting the conclusions of this article will be made available by the authors, without undue reservation.

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
