# Peer review of "Understanding Hand Hygiene Behavior in a Public Hospital in Benin Using the Theoretical Domain Frameworks: The First Step for Designing Appropriate Interventions"

_healthcare, 2022, doi:10.3390/healthcare10101924_

Round 1

Reviewer 1 Report

Nosocomial infections are a major cause of morbidity and mortality in hospitalized patients.The most effective and inexpensive way to prevent these infections is to increase the compliance of hospital staff with hand hygiene. The aim of the study were  to reveal the behavior of hospital staff on hand hygiene and to develop suggestions for improvement. The strengths of this study is the  use of the TDF to determine the hand hygiene behavior of hospital staff. The manuscript is clear, relevant for the field and presented in a well-structured.The cited references are mostly recent publication.The results were interpreted appropriately.

Author Response

Dear Reviewer 1, 

Thank you for your comments and good advices regarding our manuscript. 

We correct the article according to the comments of other two reviewers. 

regards 

Carine 

Reviewer 2 Report

Thank you for sharing this interesting manuscript titled, Understanding Hand Hygiene Behavior in A Public Hospital in 2 Benin Using the Theoretical Domain Frameworks. The manuscript presents views of healthcare workers (HCWs) on HH behavior in public hospital in Benin by using semi-structured interviews. This method and framework is effective and has been used at other settings.

The authors can use experiences from other published articles- for example,

1.      “How can the patients remain safe, if we are not safe and protected from the infections”? A qualitative exploration among health care workers about challenges of maintaining hospital cleanliness in a resource limited tertiary setting in rural India. Sudhir Chandra Joshi, Vishal Diwan, Rita Joshi, Megha Sharma, Ashish Pathak, Harshada Shah, Ashok J Tamhankar, Cecilia StålsbyLundborg. IJERPH. Sept 2018.

2.      A step-wise approach towards introduction of an alcohol based hand rub, and implementation of front line ownership- using a, rural, tertiary care hospital in central India as a model. Sharma M*, Joshi R, Shah H, Macaden R and StålsbyLundborg C. BMC Health Serv Res. 2015 Apr 29;15(1):182. DOI: 10.1186/s12913-015-0840-1. PMID: 25924956 Published 29 April 2015

3.      Understanding healthcare workers self-reported practices, knowledge and attitude about hand hygiene in a medical setting in rural India. Diwan V, Gustafsson C, Rosales-Klintz S, Joshi SC, Joshi R, Sharma M, Shah H, Pathak A, Tamhankar AJ, StalsbyLundborg C, PLoS ONE, Sept 2016.

Please find below some more comments-

1.       Did the HCWs know who was taking their interviews? How comfortable were the HCWs to give an interview to an entirely new person?

2.       Will you recommend implementation of this method (Interview by a new person) at other settings or do you find this as a limitation? Do you expect to get more information if the interviewes were taken by someone in the system?

3.       What was the language of interview?

4.       It is not mentioned that how many HCWs were working in total in the critical point care? Were all available HCWs included in the study.

5.       Was there any correlation of factors affecting hand hygiene and sex of the participants?

6.        Line 311- “Some interviewees have challenged us about how dry they hands became because of 311 HH practice in work.” What does they meant here? I think it is misspelt.

7.       Participants indicated about dry hands due to use of ABHR however- an appropriate ABHR must include an emollient to soothe skin of hands. What type of ABHR was used in the setting?

8.       Participants also doubted the quality of water. Was there any study conducted to check this point? Have you planned any such study for future?

9.       What was the SSI rate in the setting?

10.   There are some issues in transcribing or maybe it is in English, for example- please see this sentences in Table 2. “Sometimes I wash my hands, but without efficacity because water is non clean. When I washed also hands without soap and if my technique isn’t good, I’m not compliant” (Interview 1, nurse).”

11.    Participants suggested regular and repeated training as an important factor as well as chances of inappropriate hand washing (social or not as per WHO’s suggested steps of hand hygiene). However, only some participants received training, it is not clear how many?.

12.   Line 381- “Moreover, in our setting, hand hygiene topic is not included in 381 training curriculum.”, which training curriculumn? What kind of trainings were provided at the settings? What was the reason to exclude Hand hygiene from the curriculum?

13.   The results are important and need to be addressed. Please explain What are the future implications of the results.

Few General comments

14.   Remove extra spaces in between words.

15.   Some of the quotes are not italicized.

16.   Extensive checking for English language is suggested

Author Response

Dear reviewer 2 , 

Thank for your constructive comments. 

Please find the point by point response to your comments 

Regards 

Reviewer 3 Report

I found this article very interesting and well constructed. I made some comments to improve the understanding of the methodology directly in the PDF.

Author Response

Dear reviewer 3, 

Thank for your constructive remarks.

Please find the response to your comments. 

Regards 

carine 

Round 2

Reviewer 2 Report

Thank you for the responses. The responses to the comments are satisfactory and the manuscript can be accepted after thorough checking of the language.